# Combined Host- and Pathogen-Directed Therapy for the Control of *Mycobacterium abscessus* Infection

Noemi Poerio,[a] Camilla Riva,[b] Tommaso Olimpieri,[a] Marco Rossi,[b] Nicola I. Lorè,[b] Federica De Santis,[a] Lucia Henrici De Angelis,[a] Fabiana Ciciriello,[c] Marco M. D'Andrea,[a] Vincenzina Lucidi,[c] Daniela M. Cirillo,[b] Maurizio Fraziano[a]

aDepartment of Biology, University of Rome "Tor Vergata", Rome, Italy
bEmerging Bacteria Pathogens Unit, San Raffaele Scientific Institute, Milan, Italy
cDepartment of Pediatric Medicine, Cystic Fibrosis Complex Operating Unit, Bambino Gesù Pediatric Hospital, Rome, Italy

Noemi Poerio and Camilla Riva contributed equally to this article. Author order was determined by their equal but gradated contributions to this paper. Maurizio Fraziano and Daniela M. Cirillo contributed equally to this article; the author order was determined by their equal but gradated contributions to this paper.

**ABSTRACT** *Mycobacterium abscessus* is the etiological agent of severe pulmonary infections in vulnerable patients, such as those with cystic fibrosis (CF), where it represents a relevant cause of morbidity and mortality. Treatment of pulmonary infections caused by *M. abscessus* remains extremely difficult, as this species is resistant to most classes of antibiotics, including macrolides, aminoglycosides, rifamycins, tetracyclines, and $\beta$-lactams. Here, we show that apoptotic body like liposomes loaded with phosphatidylinositol 5-phosphate (ABL/PI5P) enhance the antimycobacterial response, both in macrophages from healthy donors exposed to pharmacological inhibition of cystic fibrosis transmembrane conductance regulator (CFTR) and in macrophages from CF patients, by enhancing phagosome acidification and reactive oxygen species (ROS) production. The treatment with liposomes of wild-type as well as CF mice, intratracheally infected with *M. abscessus*, resulted in about a 2-log reduction of pulmonary mycobacterial burden and a significant reduction of macrophages and neutrophils in bronchoalveolar lavage fluid (BALF). Finally, the combination treatment with ABL/PI5P and amikacin, to specifically target intracellular and extracellular bacilli, resulted in a further significant reduction of both pulmonary mycobacterial burden and inflammatory response in comparison with the single treatments. These results offer the conceptual basis for a novel therapeutic regimen based on antibiotic and bioactive liposomes, used as a combined host- and pathogen-directed therapeutic strategy, aimed at the control of *M. abscessus* infection, and of related immunopathogenic responses, for which therapeutic options are still limited.

**IMPORTANCE** *Mycobacterium abscessus* is an opportunistic pathogen intrinsically resistant to many antibiotics, frequently linked to chronic pulmonary infections, and representing a relevant cause of morbidity and mortality, especially in immunocompromised patients, such as those affected by cystic fibrosis. *M. abscessus*-caused pulmonary infection treatment is extremely difficult due to its high toxicity and long-lasting regimen with life-impairing side effects and the scarce availability of new antibiotics approved for human use. In this context, there is an urgent need for the development of an alternative therapeutic strategy that aims at improving the current management of patients affected by chronic *M. abscessus* infections. Our data support the therapeutic value of a combined host- and pathogen-directed therapy as a promising approach, as an alternative to single treatments, to simultaneously target intracellular and extracellular pathogens and improve the clinical management of patients infected with multidrug-resistant pathogens such as *M. abscessus*.

**KEYWORDS** chronic infection, cystic fibrosis, host-pathogen interactions, infectious disease, innate immunity, liposomes, nontuberculous mycobacteria, pulmonary infection

Address correspondence to Daniela M. Cirillo, cirillo.daniela@hsr.it, or Maurizio Fraziano, fraziano@bio.uniroma2.it.
The authors declare no conflict of interest.

Cystic fibrosis (CF) is an autosomal recessive genetic disease caused by mutations in the gene encoding the cystic fibrosis transmembrane conductance regulator (CFTR) channel, resulting in a systemic disease which affects different organs and apparatuses, such as the pancreas, liver, reproductive tract, and mainly, the lungs (1). Here, the loss of function of CFTR causes a defective mucociliary clearance, a dramatic production of sticky mucus, and dysfunction in the phagocytosis process (2, 3), leading to chronic bacterial infection and colonization by opportunistic pathogens, such as *Mycobacterium abscessus* (4). *M. abscessus* represents a relevant cause of morbidity and mortality in CF patients, where it causes severe pulmonary infections, which are very difficult to treat due to its trend to form biofilms and its natural resistance to a wide range of antibiotics (5). On these grounds, the identification of novel therapeutic strategies capable of supporting or complementing the few currently available antibiotic options represents an urgent issue to be addressed.

Phagocytosis is a key effector mechanism of the antimicrobial innate immune response and is mediated by a timely coordinated and a topologically distributed expression of second lipid messengers, which recruit signal proteins on the nascent or maturing phagosome through recognition of specific lipid-binding domains (6, 7). We have previously demonstrated that selected second lipid messengers, involved in phagolysosome biogenesis and delivered by apoptotic body-like liposomes (ABLs), enhance innate (myco)bactericidal responses in different *in vitro*, *ex vivo*, and *in vivo* experimental models of pathogen-inhibited and host-impaired phagosome maturation (8, 9), as well as simultaneously reducing potentially damaging tissue inflammatory responses (9, 10). In particular, the treatment with ABLs loaded with phosphatidic acid (PA) of *Mycobacterium tuberculosis*-infected mice resulted in a 100-fold reduction of pulmonary mycobacterial loads, with a concomitant 10-fold reduction of serum tumor necrosis factor alpha (TNF-$\alpha$), interleukin-1beta (IL-1$\beta$), and gamma interferon (IFN-$\gamma$) (9). Furthermore, ABLs loaded with phosphatidylinositol 5-phosphate (PI5P) improve the antimicrobial response to multidrug-resistant (MDR) *Pseudomonas aeruginosa* in impaired macrophages from CF patients and limit *in vivo* airway inflammatory response (10). Therefore, we evaluated ABLs loaded with selected bioactive lipids, alone or in combination with amikacin, as a novel formulation of a combined host- and pathogen-directed therapeutic approach against *in vitro* and *in vivo M. abscessus* infection.

## RESULTS

**ABLs loaded with PA, PI3P, and PI5P improve *M. abscessus* uptake and intracellular killing both in dTHP-1 cells and monocyte-derived macrophages (MDM) treated or not treated with INH172.** The bioactive lipids PA, PI5P, phosphatidylinositol 3-phosphate (PI3P), lysobisphosphatidic acid (LBPA), sphingosine 1-phosphate (S1P), and arachidonic acid (AA), chosen because of their capability to drive the phagocytosis process (6, 7), were individually included in the inner layer of ABLs (see Fig. S1A in the supplemental material) to test their *in vitro* capability to enhance intracellular mycobacterial killing. These liposome formulations were preliminarily tested in flow cytometry for size distribution by comparing their forward scatter parameters (FS) with the FS of commercially available beads of known diameter. Results show a size distribution of the different liposome formulations between 0.8 $\mu$m and 6 $\mu$m in diameter (Fig. S1B to G), which indicates their suitability to be used for inhalation against pulmonary infections (11). The different liposome formulations were then used to stimulate differentiated THP1 (dTHP-1) cells infected with *M. abscessus* and were evaluated in terms of mycobacterial phagocytosis and replication index. Results show that liposomes loaded with PA (ABL/PA), PI3P (ABL/PI3P), and PI5P (ABL/PI5P) enhanced internalization of *M. abscessus* in dTHP-1 cells treated (Fig. 1A) or not treated (Fig. S2A) with the pharmacological inhibitor of CFTR, INH172. After mycobacterial internalization, phagosome acidification and reactive oxygen species (ROS) generation are sequential steps responsible for the killing of intracellular pathogens (12). Macrophages exposed to the same three liposome formulations promote phagosome acidification (Fig. S3A) and ROS generation (Fig. S3B). To measure the functional consequences of phagosome acidification and of ROS generation, we measured intracellular *M. abscessus* viability following stimulation with all liposome formulations. The analysis confirmed that ABL/PA, ABL/PI3P, and ABL/PI5P, among six liposome formulations, significantly reduced the mycobacterial replication index in dTHP-1 (Fig. S2B) and in

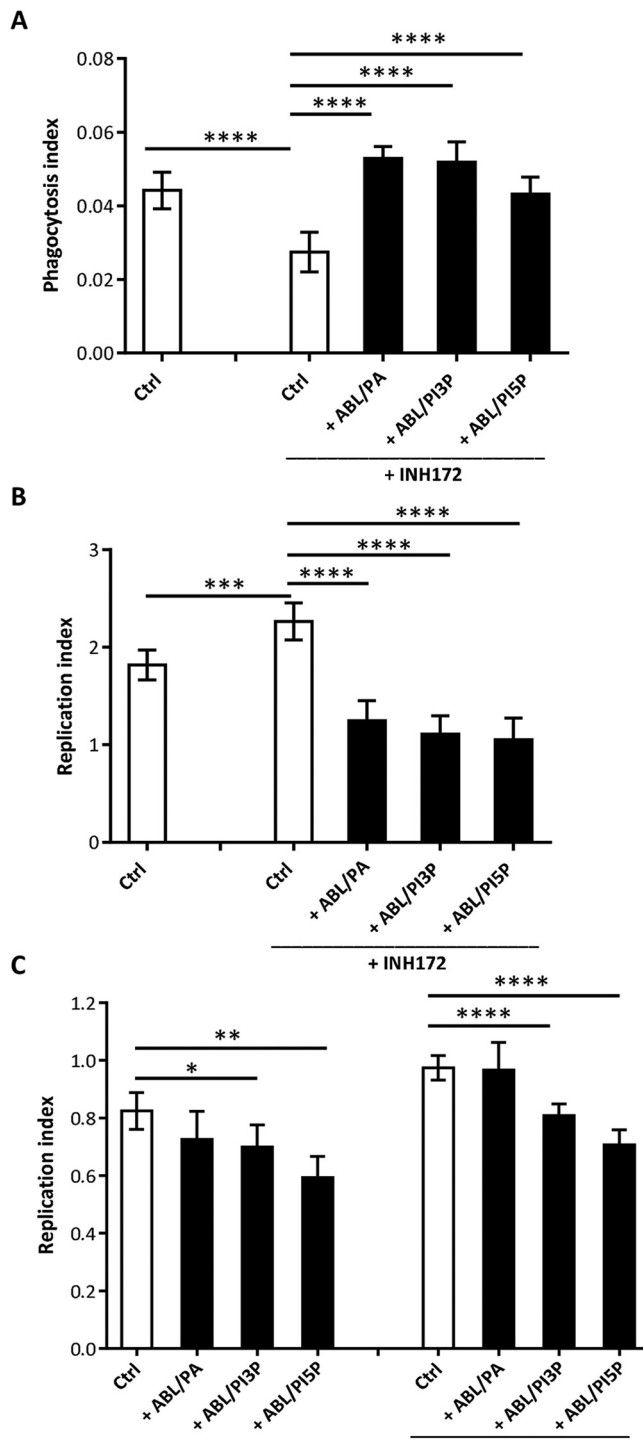

**FIG 1** ABL/PA, ABL/PI3P, and ABL/PI5P enhance both internalization and intracellular killing of *M. abscessus* in dTHP-1 cells and monocyte-derived macrophages (MDM) from healthy donors (HD) with pharmacologically inhibited CFTR. (A to C) dTHP-1 cells (A and B) and primary MDM from healthy donors (C), treated or not treated with INH172, were cultured at $5 \times 10^5$ cells/well in 24-well plates and $2 \times 10^5$ cells/well in 96-well plates. (A) Cells were stimulated with selected ABL formulations before infection for 30 min and then infected with the *M. abscessus* reference strain (ATCC 19977). Bacterial uptake was quantified by CFU assay and indicated as phagocytosis index, calculated as the ratio between the CFU obtained immediately after the infection and those from the inoculum. (B and C) Cells, dTHP-1 cells (B) and MDM from healthy donors (C), were exposed or not exposed to INH172, infected with *M. Abscessus*, and treated for 18 h with the selected ABL formulations. Bacterial growth was assessed by CFU assay. The replication index was calculated as the ratio between the CFU

dTHP-1 following pharmacological inhibition of CFTR (Fig. 1B). These results were further confirmed in primary macrophages from healthy donors (Fig. 1C).

**ABL/PI5P promotes intracellular *M. abscessus* killing dependent on ROS production and phagosome acidification in CF macrophages.** Previous results induced us to initially select ABL/PA, ABL/PI3P, and ABL/PI5P for subsequent analyses on primary monocyte-derived macrophages isolated from CF patients. Macrophages from CF patients were infected *in vitro* with *M. abscessus*, and bacterial viability was measured by CFU assay and shown as replication index. Results show that no significant effect was observed following stimulation with ABL/PA (Fig. S4A), whereas ABL/PI3P and, most significantly, ABL/PI5P induced a significant improvement of antimycobacterial activity compared to unstimulated cells (Fig. S4B and Fig. 2A), indicating that the effect was specific and dependent upon the second lipid messenger used. Given that ABL/PI5P resulted in the most significant effect, we next investigated its antimicrobial activity mechanism, observing that intracellular mycobacterial killing was dependent on phagosome acidification and ROS generation. In particular, CF macrophages were exposed to concanamycin A, an inhibitor of V-ATPase (13), or to pegylated catalase (PEG-Cat), which converts hydrogen peroxide to water and oxygen and thus reduces ROS activity. Results indicate that concanamycin A (Fig. 2B) and PEG-Cat (Fig. 2C) almost completely abolish the ABL-induced intracellular *M. abscessus* killing, demonstrating that the main mechanism of ABL/PI5P action relies on phagosome acidification and ROS generation.

**ABL/PI5P induces *M. abscessus* clearance and leukocyte recruitment in wild-type (WT) and CF mice.** Next, we tested the therapeutic value of the treatment with ABL/PA, ABL/PI3P, and ABL/PI5P in an *in vivo* model of *M. abscessus* infection. To this aim, immunocompetent C57BL/6 mice were intratracheally (i.t.) infected with *M. abscessus* (reference strain ATCC 19977), and a chronic lung infection was established by a previously described agar bead method (14). After 1 week from *M. abscessus* infection, mice were treated 3 times per week for 4 consecutive weeks by intranasal inoculation of $10^5$ ABLs loaded with PA, PI3P, or PI5P. After 8, 17, and 29 days of treatment, mice were sacrificed, bronchoalveolar lavage (BAL) was performed, and the total lung was processed for microbiological analysis and for the assessment of inflammatory parameters (Fig. 3A). Results showed that all ABL formulations induced a progressive reduction of pulmonary mycobacterial burden over time, reaching about a 2-log reduction of mean pulmonary CFU after 29 days of treatment. Notably, after 29 days of treatment with ABL/PA and ABL/PI5P, in 3 out of 12 mice the total lung CFU was not detectable by the viable count assay (Fig. 3B). The analysis of the inflammatory infiltrate revealed a strongly significant reduction of neutrophils in bronchoalveolar lavage fluid (BALF) of mice treated with all ABL formulations at all time points analyzed and a less pronounced, but still significant, reduction of macrophages in BALF of mice treated with ABL/PA and ABL/PI3P for 8 and 29 days (Fig. 3C). Such a reduction of inflammatory cells in the lungs of treated mice was associated, only in the case of ABL/PI5P treatment, with a progressive decrease of keratinocyte-derived chemokine (KC) (albeit not significant) and a significant decrease of both IL-1$\beta$ and IFN-$\gamma$ after 8 and 29 days of treatment, whereas TNF-$\alpha$ and IL-6 did not show any significant modulation over time (Fig. S5). Based on the above-described *in vitro* and *in vivo* results, and considering the crucial role played by the inflammatory response in the pathogenesis of CF (15), we selected ABL/PI5P for subsequent *in vivo* analysis on CF mice. In particular, CFTR knockout (KO) and wild-type mice were infected with *M. abscessus* and, starting from day 8 after infection, treated 3 times per week for 4 consecutive weeks by intranasal inoculation of $10^5$ ABLs carrying PI5P. After 8 and 29 days of treatment, mice were sacrificed, BALF was recovered, and total the lung was processed for cell counts and CFU analysis, respectively (Fig. 4A). Results showed an about 2-log reduction of pulmonary bacterial burden following liposome administration both in wild-type and

**FIG 1** Legend (Continued)
obtained 18 h after infection, in the presence or absence of ABL formulations, and the CFU obtained before the addition of liposomes. The results are shown as the mean ± standard deviation of the values obtained from triplicates of each condition, and panel C is representative of experiments with cells from four different donors. *, $P < 0.05$; **, $P < 0.01$; ***, $P < 0.001$; ****, $P < 0.0001$ by Student's *t* test.

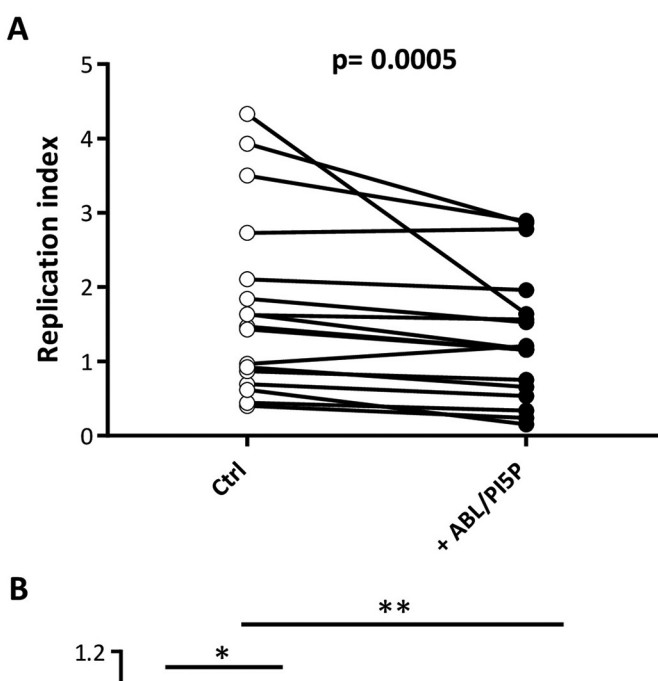

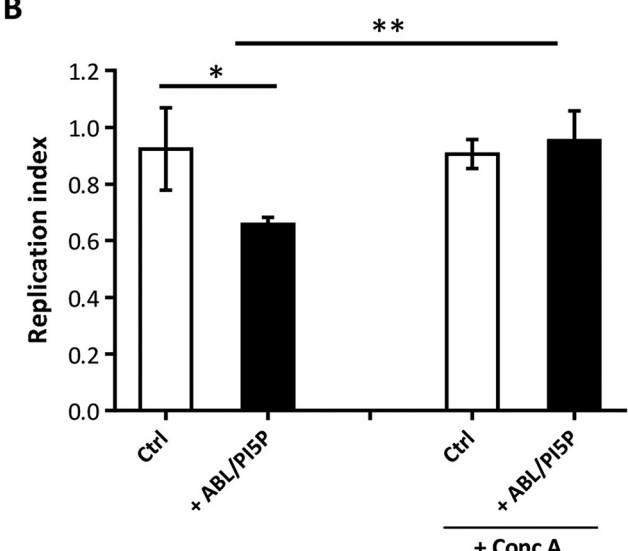

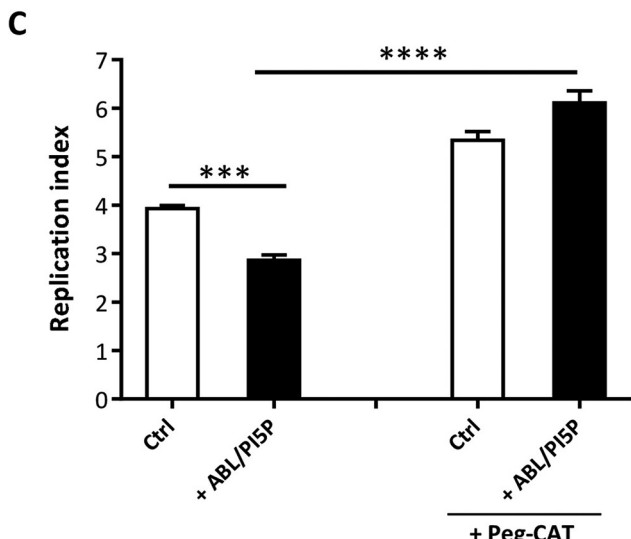

**FIG 2** ABL/PI5P promotes both ROS- and phagolysosome acidification-dependent intracellular *M. abscessus* killing in CF MDM. (A to C) MDM isolated from CF patients (*n* = 17) were plated at the

CFTR KO mice treated for 8 and 29 days (Fig. 4B), which was associated with a significant reduction of total cell counts in BALF (Fig. 4C).

**ABL/PI5P-amikacin combined treatment induces both clearance and anti-inflammatory responses in CF MDM and in an *in vivo* model of *M. abscessus* infection.** As a combination therapy may represent a valuable strategy to differentially target extracellular and intracellular pathogens, we combined ABL/PI5P with amikacin antibiotic treatment, chosen for its clinical use in both intensive and continuation therapeutic regimens for *M. abscessus* infection (16) and for its low penetration capability in macrophages (17). Initially the combined treatment was tested *in vitro* on *M. abscessus*-infected dTHP-1 cells treated or not treated with CFTR inhibitor (INH172) as well as on macrophages from CF patients. Next, we evaluated, in an *in vivo* model of chronic *M. abscessus* infection, the efficacy of the combined strategy with administration of intranasal (i.n.) ABL/PI5P and intraperitoneal (i.p.) amikacin.

Results show that ABL/PI5P reduces the intracellular *M. abscessus* replication in dTHP-1 cells irrespective of CFTR inhibition (Fig. S6B and D), without any direct effect on the extracellular pathogen (Fig. S6A and C). The combination treatment with ABL/PI5P and AMK (Fig. S6B and D) induced a significantly higher reduction of *M. abscessus* replication index than the single treatments. Similar analysis was then performed on monocyte-derived macrophages from CF patients, and results confirmed the increased therapeutic value of the *in vitro* combined treatment, compared to the single treatments, on extracellular and intracellular pathogens (Fig. 5A and B). Such a combination of host-directed (ABL/PI5P) and pathogen-directed (amikacin) therapy, was finally tested on wild-type C57BL/6 mice intratracheally infected with *M. abscessus*. Mice were intratracheally infected with *M. abscessus* (reference strain ATCC 19977) and, starting on day 7 after infection, were inoculated daily, via the i.p. route, with 100 mg/kg body weight of amikacin, whose dose was chosen in preliminary *in vivo* experiments (Fig. S7) and/or treated 3 times per week for 4 consecutive weeks by intranasal inoculation of $10^5$ liposomes. After 8 and 29 days of treatment mice were sacrificed, BALF was obtained, and the total lung was processed for microbiological analysis and assessment of inflammatory parameters (Fig. 6A). Results show that the combination therapy induced a highly significant decrease in pulmonary mycobacterial burden compared to the single therapies (Fig. 6B) along with a significant reduction of neutrophils and macrophages in BALF (Fig. 6C), in the absence of any signs of kidney and liver toxicity (Fig. S8). The reduction of inflammatory response was also evaluated in terms of cytokines in BALF and in total lung. Results obtained after 29 days of treatment show that the combination therapy induced a more significant reduction of IL-1$\beta$, TNF-$\alpha$, KC, IL-17a, and IFN-$\gamma$ in total lung than the single treatments (Fig. S9).

The combination of host- and pathogen-directed therapy was finally tested in CFTR KO mice, a relevant experimental model of CF, after *M. abscessus* infection (Fig. 7A). Results show that a significant reduction of pulmonary mycobacterial burden (Fig. 7B) and inflammatory response (Fig. 7C) was observed following combination therapy.

## DISCUSSION

The therapeutic management of *M. abscessus* infection remains extremely difficult, as this pathogen exhibits resistance to many different antibiotics (18). In addition to mutations in drug targets, resistance mechanisms may involve (i) intrinsic drug resistance,

**FIG 2** Legend (Continued)
concentration of $1 \times 10^6$ cells/mL, infected with the *M. abscessus* reference strain (ATCC 19977), and then stimulated for 18 h with ABL carrying PI5P (A) (*n* = 17) in the presence or absence of catalase (PEG-Cat) (B) (representative of *n* = 4) or concanamycin A (Conc A), a specific inhibitor of vacuolar type H$^+$-ATPase activity (C) (representative of *n* = 4) at the concentration of 100 U/mL or 1 nM, respectively. Bacterial growth was assessed by CFU assay, and the replication index was calculated as the ratio between the CFU obtained 18 h after infection, in the presence or absence of ABL formulations, and the CFU obtained before the addition of liposomes. (A) Statistical analysis was performed using the two-sided Wilcoxon matched-pair signed rank test (*P* = 0.0005). (B and C) The results are shown as the mean ± standard deviation of the values obtained from each condition and are representative of experiments with cells from four different CF patients. *, *P* < 0.05; **, *P* < 0.01; ***, *P* < 0.001; ****, *P* < 0.0001 by Student's *t* test.

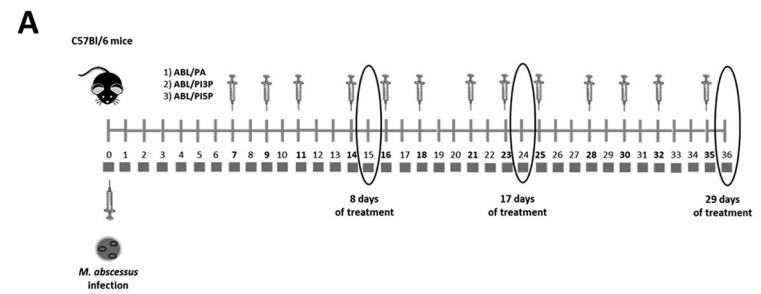

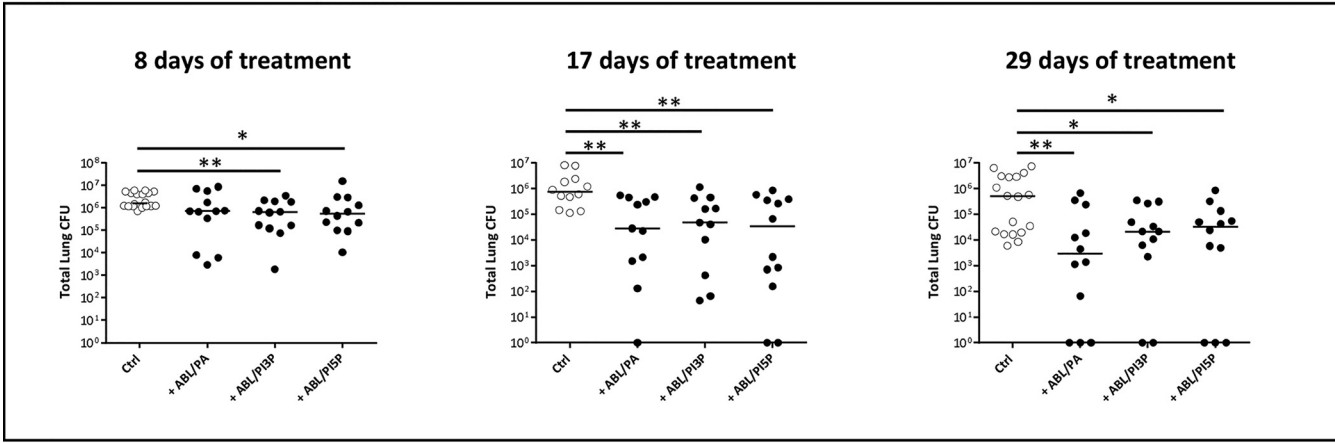

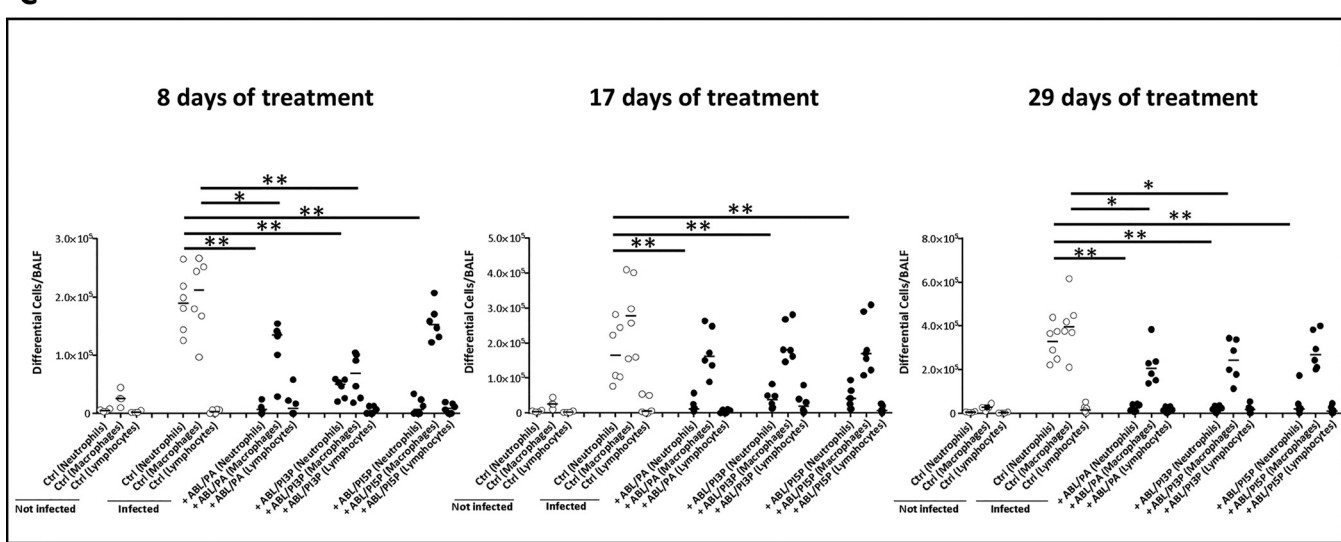

**FIG 3** ABL/PI5P reduces both bacterial burden and leukocyte recruitment in WT mice infected with *M. abscessus*. (A) C57BL/6N mice were chronically infected with $10^5$ CFU of the *M. abscessus* reference strain (ATCC 19977) by intratracheal (i.t.) injection and treated after 1 week of infection with ABL loaded with PA, PI3P, or PI5P three times a week for 8, 17, and 29 days. (A and B) After these time points mice were sacrificed, and BALF and lungs were processed for microbiological analysis (B) and evaluation of neutrophil, macrophage, and lymphocyte BALF recruitment (C). Data from two independent experiments were pooled. The results are shown as the median of the values, and statistical significance by Mann-Whitney test was determined (*, $P < 0.05$; **, $P < 0.01$).

(ii) low permeability of the cell wall, (iii) induction of drug efflux pumps, (iv) mutations in mycobacterial enzymes which do not convert prodrugs into active metabolites, and/or (v) the expression of numerous enzymes that can either neutralize drugs or modify their specific targets (5). The standard therapy of pulmonary *M. abscessus* infection in CF patients requires an intensive phase and a continuation phase based on different therapeutic regimens, whose duration depends upon the severity of infection and upon the response to treatment (19). In particular, the intensive phase consists of a daily oral

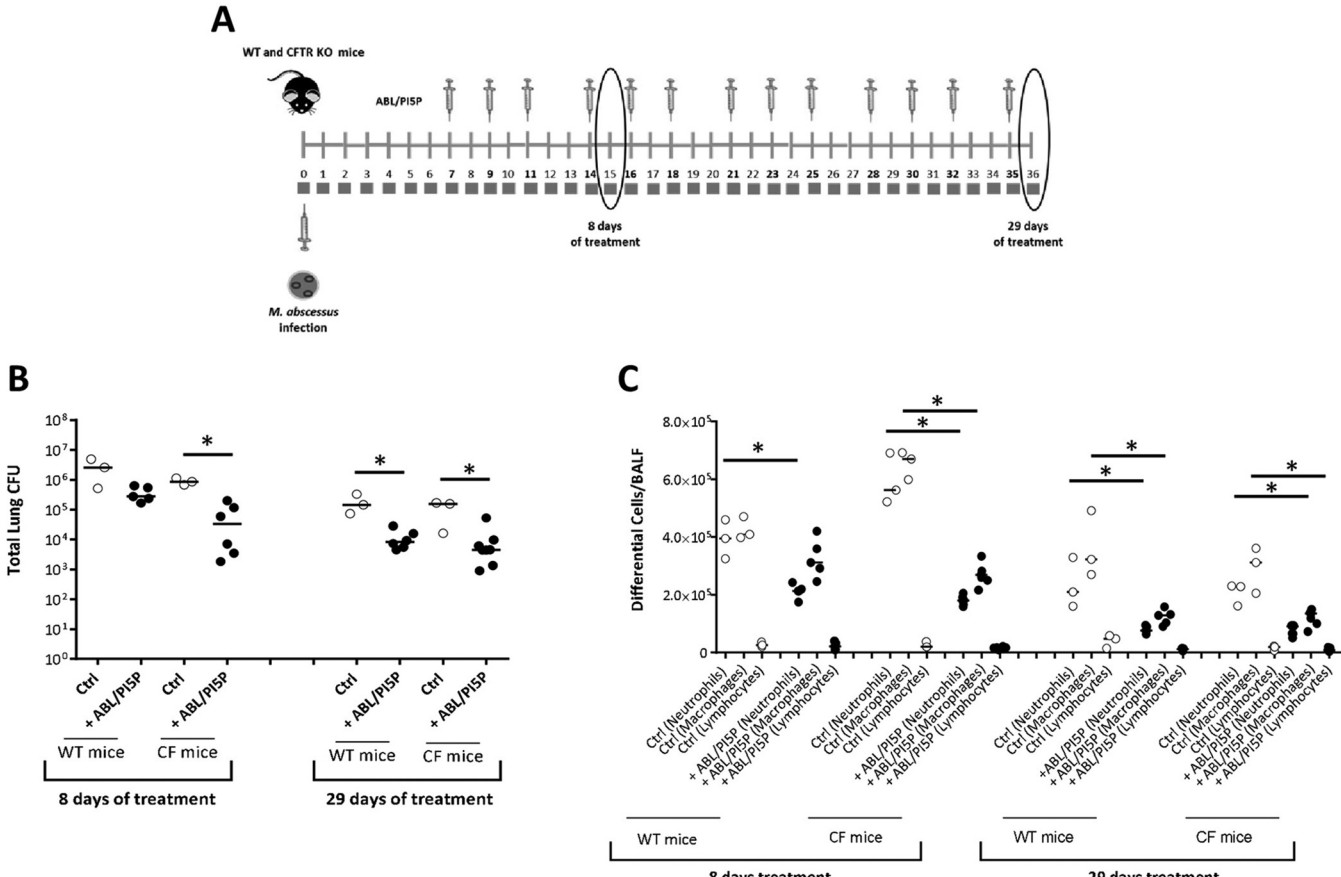

**FIG 4** ABL/PI5P reduces both bacterial burden and leukocyte recruitment in CF mice infected with *M. abscessus*. (A) WT and CF mice were chronically infected with 10^5 CFU of the *M. abscessus* reference strain (ATCC 19977) by i.t. injection and treated with ABL/PI5P three times a week for 8 and 29 days. (B and C) Mice were then sacrificed, and BALF and lungs were processed for (B) microbiological analysis and (C) evaluation of neutrophil, macrophage, and lymphocyte BALF recruitment to establish the therapeutic efficacy of liposomes. The results are shown as the median of the values, and statistical significance by Mann-Whitney test was determined (*, $P < 0.05$; **, $P < 0.01$).

administration of a macrolide together with the intravenous inoculation of amikacin and one or more among tigecycline, imipenem, or cefoxitin. The continuation phase is based on daily oral administration of a macrolide in association with inhaled amikacin together with 2 or 3 among minocycline, clofazimine, moxifloxacin, and linezolid (16). Such a complex therapeutic regimen is worsened by drug intolerance and drug-related toxicity, frequently occurring in CF patients, often requiring changes in antibiotic therapy (16). Thus, the discovery of additional and more effective anti-*M. abscessus* drugs and/or the identification of novel therapeutic strategies are urgently needed for the clinical management of this pathogen.

CF macrophages show signs of impaired phagolysosome maturation, due to the CFTR mutation-dependent block of phagosomal acidification, which leads to a reduced antimicrobial response (3, 10). In addition, *M. abscessus* has been reported to further inhibit phagosome maturation and to escape from the phagosome microenvironment by the ESX-4 system (20), which may further facilitate colonization of vulnerable CF patients. Thus, an efficient host-directed therapy aimed to enhance intracellular mycobacterial killing should rescue both the pathogen-inhibited and host-impaired phagosome maturation process. In the present study, we show *in vitro* and *in vivo* results supporting the therapeutic value of the combined treatment with ABL/PI5P and amikacin, used as host- and pathogen-directed therapy, aimed at specifically targeting the intracellular and extracellular pathogen, respectively.

The role of PI5P in the activation of innate antimicrobial responses has been reported in different kinds of infection. In particular, we have recently reported that ABL loaded with PI5P could enhance intracellular killing of *P. aeruginosa* in impaired macrophages from CF patients (10) and of *in vivo* acquired *Klebsiella pneumoniae*, *Klebsiella oxytoca*, *Acinetobacter*

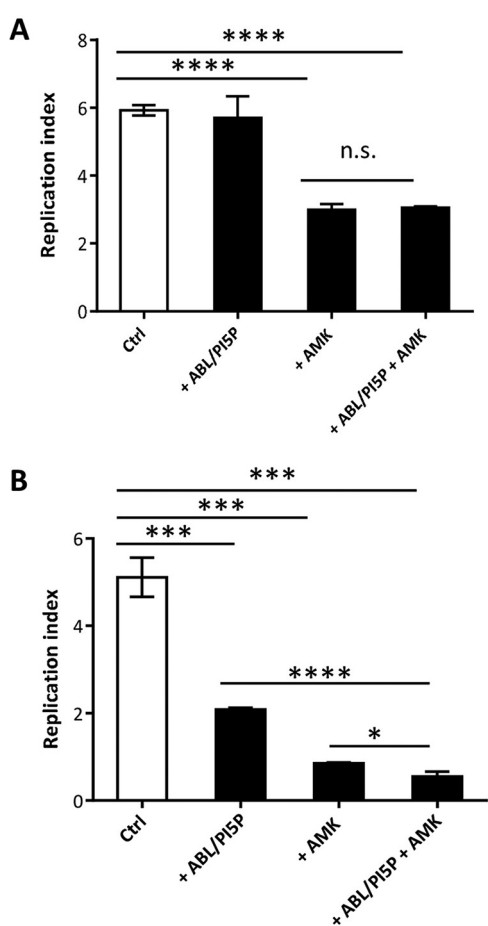

**FIG 5** ABL/PI5P-amikacin combined treatment promotes higher reduction of *M. abscessus* intracellular growth than single treatments in CF MDM. CF MDM were cultured and infected as described in Materials and Methods. (A and B) Finally, supernatant was collected, cells were lysed, and both were analyzed for extracellular (A) and intracellular (B) bacterial growth. The replication index was calculated as the ratio between the CFU obtained 18 h after infection in the presence or absence of ABL/PI5P and/or amikacin (AMK) and those obtained immediately after infection, before the addition of the stimuli. Results are shown as the mean $\pm$ standard deviation of the values obtained from triplicates of each condition and are representative of four CF patients. n.s., not significant; *, $P < 0.05$; ***, $P < 0.001$; ****, $P < 0.0001$ by Student's *t* test.

*baumannii*, and methicillin-resistant *Staphylococcus aureus* in BALF cells from patients with pneumonia caused by the respective pathogens (8). Also, the role of phosphoinositide 5-kinase (PIKfive), which catalyzes the formation of PI3,5P$_2$ and PI5P, has been recently described to play an important role in the early phases of phagosome maturation and the intracellular restriction of *Legionella pneumophila* infection (21). The results reported here further extend such evidence, indicating a very efficient capability of ABL/PI5P liposome formulation to restrict intracellular *M. abscessus* growth, by a phagosome acidification-dependent and ROS-mediated mechanism, in macrophages from healthy donors as well as CF patients. Although the exact molecular pathway activated by PI5P and leading to intracellular mycobacterial killing is still unknown, a role of PI5P in the activation of VPS34-independent noncanonical autophagy has been reported (22), which may be responsible for the observed phagosome acidification, ROS production, and intracellular mycobacterial killing.

Airway inflammation is a hallmark of CF disease characterized by elevated levels of proinflammatory cytokines and chemokines (15), resulting in chronic inflammation, neutrophil recruitment, and progressive airway destruction leading to the decline in lung function. Thus, the concomitant reduction of *in vivo* inflammatory responses may represent a further important goal for the treatment of chronic *M. abscessus* infection. The functional distribution of PS at the outer surface of cell membranes represents an "eat me" signal (23)

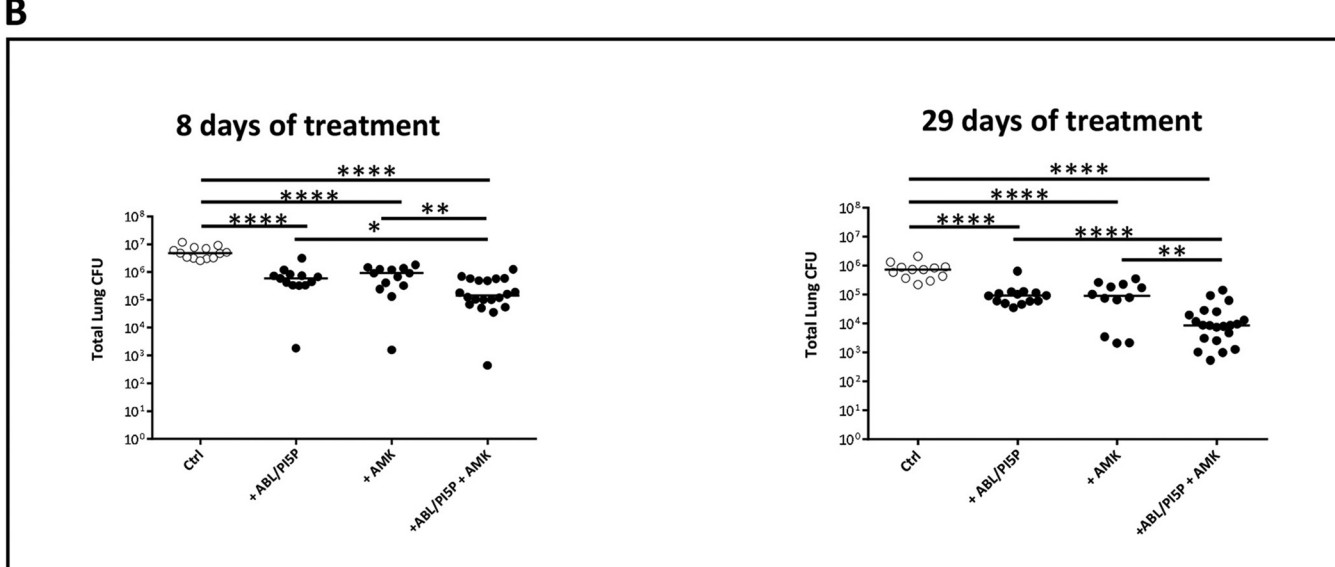

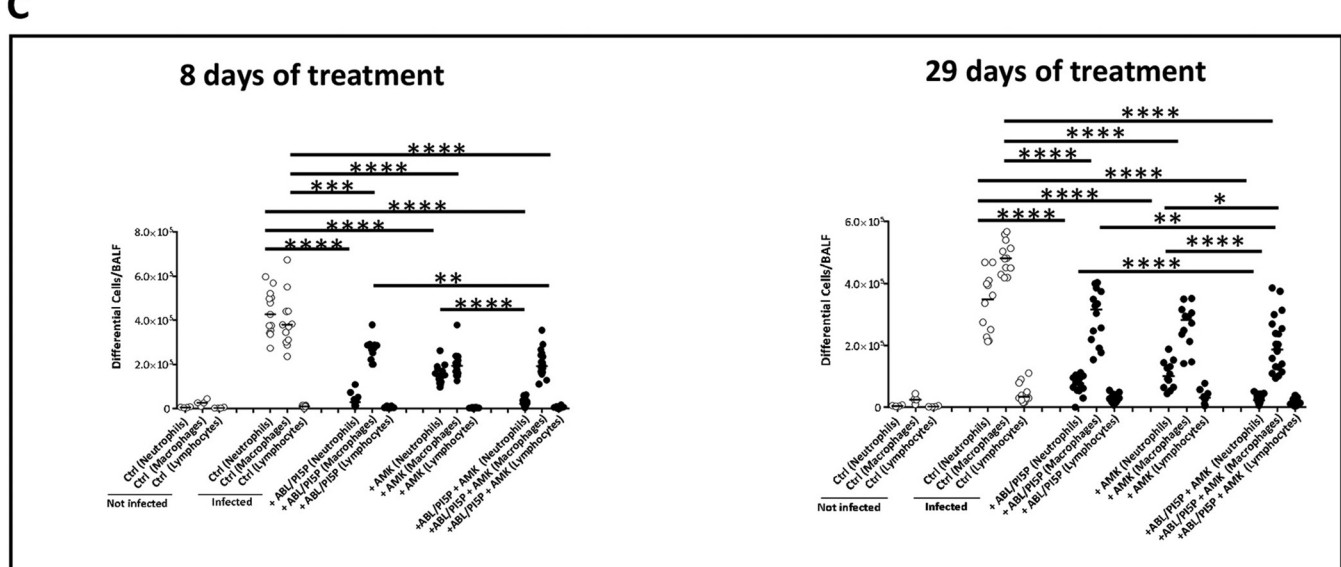

**FIG 6** ABL/PI5P-amikacin combined treatment induces both *M. abscessus* clearance and anti-inflammatory responses in WT mice. (A) WT mice were chronically infected with 10^5 CFU of the *M. abscessus* reference strain (ATCC 19977) and treated 1 week after infection with AMK and/or ABL/PI5P as described in Materials and Methods. (B and C) After 8 and 29 days of treatment, mice were sacrificed, and BALF and lungs were processed for microbiological analysis (B) and evaluation of neutrophil, macrophage, and lymphocyte BALF recruitment (C). The results are shown as the median of the values, and statistical significance by Mann-Whitney test was determined (*, $P < 0.05$; **, $P < 0.01$; ***, $P < 0.001$; ****, $P < 0.0001$).

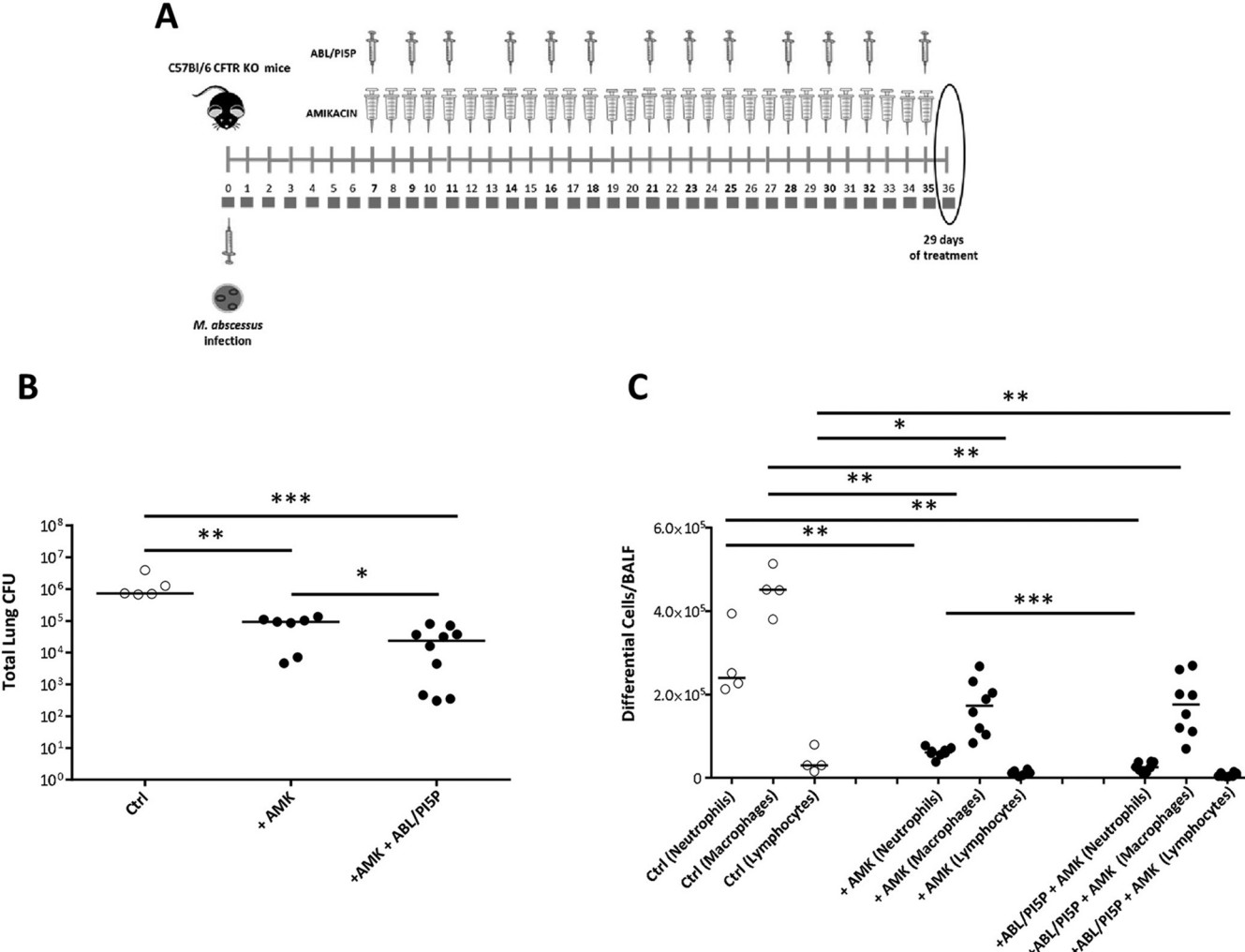

**FIG 7** ABL/PI5P-amikacin combined treatment induces both *M. abscessus* clearance and anti-inflammatory responses in CF mice. (A) CF mice were chronically infected with 10⁵ CFU of the *M. abscessus* reference strain (ATCC 19977) and treated 1 week after infection with AMK and/or ABL/PI5P as described in Materials and Methods. (B and C) After 29 days of treatment mice were sacrificed, and BALF and lungs were processed for microbiological analysis (B) and evaluation of neutrophil, macrophage, and lymphocyte BALF recruitment (C). The results are shown as the median of the values, and statistical significance by Mann-Whitney test was determined (*, $P < 0.05$; **, $P < 0.01$; ***, $P < 0.001$; ****, $P < 0.0001$).

through which apoptotic bodies are specifically eliminated by macrophages through an anti-inflammatory phagocytosis (24, 25). These features have highlighted the possibility of using apoptotic bodies and/or PS liposome formulations to manipulate the immune response for therapeutic gain by reducing immunopathologic responses (26–28). Moreover, we recently reported that the treatment with ABL loaded with PI5P reduces NF-κB activation and the downstream proinflammatory cytokine production in *in vitro P. aeruginosa*-infected macrophages and limits airway inflammation in *in vivo*-infected mice (10). Coherently with this already published data, the results reported here show a significant reduction of airway inflammation in *M. abscessus*-infected mice, which is more significant following amikacin-ABL/PI5P combination therapy, possibly as an additional effect of the antibiotic-mediated reduction of extracellular bacilli.

The main limitation of our study is based on the absence of a formal pharmacokinetic, pharmacodynamic, or toxicity assessment following the combined treatment with ABL/PI5P and amikacin. Despite the demonstration of efficacy against *in vivo* infection with *M. abscessus* and the absence of liver and renal toxicity in mice undergoing the treatment, this study is also limited by the demonstration of efficacy only in the murine models of infection. Thus, the clinical potential cannot be fully established until safety and efficacy trials of combined treatment are undertaken in humans.

In conclusion, our results show that the host- and pathogen-directed combination therapy reported here may offer the unique advantage to simultaneously target extracellular and intracellular mycobacteria. In this context, the intracellular hiding of bacterial pathogens in mammalian cells, such as phagocytes, can prevent antibiotics from killing pathogens and plays an underappreciated role in the recurrence of bacterial infections (29). The intracellular localization of a bacterial pathogen may hinder the antibiotic efficacy (i) by reducing the local active concentration of the drug due to either a poor permeability of several antibiotics to the cell membrane and/or a higher efflux/influx ratio, (ii) by activating the expression of genes associated with latency and by reducing the bacterial metabolism, both necessary to survive inside a hostile intracellular microenvironment (30), and (iii) by colonizing intracellular compartments that are difficult for the drug to reach (31). These persisting intracellular mycobacteria that are difficult for the drug to reach are crucial for the establishment of chronic infection and granuloma formation (5). Thus, the higher efficiency of host- and pathogen-directed combination therapy in treating *in vivo* infections by targeting intracellular and extracellular mycobacteria may translate to a reduction of the time of therapy and to minimizing the risk of emergence of additional drug resistances.

## MATERIALS AND METHODS

**Liposome preparation.** ABLs were produced as previously described (9). Briefly, the inner monolayer lipids (0.05 mg/mL) were suspended in anhydrous dodecane (Sigma). The following bioactive lipids were used for the inner monolayer: L-$\alpha$-phosphatidic acid (PA), 1,2-dioleoyl-*sn*-glycero-3-phospho-(1′-myo-inositol-3′-phosphate) (PI3P), 1,2-dioleoyl-*sn*-glycero-3-phospho-(1′-myo-inositol-5′-phosphate) (PI5P), D-erythro-sphingosine-1-phosphate (S1P), lysobisphosphatidic acid (LBPA), (all from Avanti Polar Lipids), and arachidonic acid (AA) (Sigma). L-$\alpha$-phosphatidylserine (PS) (Avanti Polar Lipids) was used as the outer monolayer lipid and was added to a 99:1 dodecane:silicone solution to obtain a final concentration of 0.05 mg/mL. Asymmetric liposomes were prepared by adding 2 mL of outer monolayer lipid suspension over 3 mL of complete medium. Finally, 100 $\mu$L of the inner monolayer lipid suspensions were added over 2 mL of lipid phase, and the samples were centrifuged at 120 $\times$ *g* for 10 min. After the centrifugation, ABLs were collected in an aqueous phase using a 5-mL syringe with a 16-gauge stainless steel needle. The following six ABL formulations were produced: PS outside/PA inside (ABL/PA), PS outside/PI3P inside (ABL/PI3P), PS outside/PI5P inside (ABL/PI5P), PS outside/S1P inside (ABL/S1P), PS outside/LBPA inside (ABL/LBPA), and PS outside/AA inside (ABL/AA). Liposomes were then quantified and analyzed in terms of dimension by two different flow cytometers, FACSCalibur (Becton, Dickinson) and FACSCelesta (Becton, Dickinson).

**Cell culture.** The human promonocytic THP-1 leukemia cell line was supplied by the European Collection of Cell Culture, grown in RPMI 1640 containing fetal bovine serum (10%), gentamicin (5 $\mu$g/mL), L-glutamine (2 mM), nonessential amino acids (1 mM), and sodium pyruvate (1 mM) and cultured in 75-cm$^2$ polystyrene flasks. Before the experiments, cells (5 $\times$ 10$^5$ or 1 $\times$ 10$^6$ per well) were seeded in 24-well plates, and the cells were induced to differentiate by stimulation for 72 h with phorbol 12-myristate 13-acetate (PMA) (20 ng/mL) and used as a model of human macrophages (dTHP-1).

Primary monocyte-derived macrophages (MDM) were prepared as previously described (10). Briefly, peripheral blood mononuclear cells (PBMC) from healthy donors and CF patients were isolated by Ficoll density gradient, and the monocytes were then positively sorted using anti-CD14 monoclonal antibodies conjugated to magnetic microbeads (Miltenyi Biotec) according to manufacturer's instructions. The monocytes were then suspended in complete medium and incubated for a further 5 days in 96-well plates at the concentration of 10$^6$ cells/mL in the presence of macrophage colony-stimulating factor (M-CSF) (50 ng/mL) (Miltenyi Biotec) to obtain differentiated macrophages.

**Bacteria.** *Mycobacterium abscessus* (ATCC 19977) was used in *in vitro* experiments and in the *in vivo* mouse model of chronic *M. abscessus* infection (14). The *M. abscessus* single colonies were collected by streaking on Middlebrook 7H10 medium (7H10; BD Difco) supplemented with oleic acid, albumin, dextrose, and catalase (OADC), then suspended in 15 mL of Middlebrook 7H9 broth (7H9; BD Difco) supplemented with albumin, dextrose, and catalase (ADC), and grown in an Erlenmeyer flask at 37°C with stirring for 48 h. The growth of bacterial cultures was monitored by measuring the optical density at the wavelength of 600 nm by a spectrophotometer (Varioskan LUX multimode microplate reader; Thermo Fisher Scientific). Bacilli were stored at −80°C until use after suspension in the microorganism preservation system Protect (Technical Service Consultants Ltd.).

For *in vivo* experiments, *M. abscessus* (ATCC 19977) was grown for 2 days (to reach the exponential phase) in 20 mL of Middlebrook 7H9 broth. Then bacteria were embedded in agar beads, as previously described (14).

**Patients.** CF patients (*n* = 22) were enrolled at "Bambino Gesù" Children's Hospital in Rome, Italy. All of the CF patients were clinically stable at the time of blood donation (5 mL). Controls were represented by buffy coats from healthy blood donors (*n* = 4) attending at Blood Transfusion Unit of Policlinico "Tor Vergata" in Rome, Italy. Clinical and demographic features of CF patients are summarized in Table S1.

**Mouse strains.** Immunocompetent C57BL/6NCrlBR male mice (8 to 10 weeks of age) from Charles River, gut-corrected CF transmembrane conductance regulator (CFTR)-deficient male and female C57BL/6 Cftrtm1UNCTgN (FABPCFTR)#Jaw mice (CF mice), and the corresponding congenic WT mice, 10 to 17 weeks old, (originally obtained from Case Western Reserve University) were tested in the experiments. Since CF mice

had limited utility because most mice die from intestinal obstruction during the first month of life, such lethal intestinal abnormalities have been corrected by expressing the human CFTR (hCFTR) in CFTR$^{-/-}$ mice under the control of the rat intestinal fatty acid-binding protein (FABP) gene promoter to direct expression of the wild-type hCFTR complementary DNA (cDNA) to the intestinal epithelial cells of these mice. Such mice survived and showed functional correction of ileal goblet cell and crypt cell hyperplasia and cyclic AMP-stimulated chloride secretion (32). All mice were maintained in specific-pathogen-free conditions at San Raffaele Scientific Institute, Milan, Italy.

**Evaluation of *in vitro* bacterial uptake and extracellular/intracellular growth.** To assess bacterial uptake, dTHP-1 cells were distributed in 24-well plates at the concentration of $5 \times 10^5$ cells/well and were stimulated with ABL carrying PA, PI3P, PI5P, S1P, LBPA, or AA used at the ratio of 1:1 (ABL:MDM) for 30 min before infection in the presence or absence of the CFTR inhibitor INH172 (Sigma), used at the concentration of 10 $\mu$M. INH172 binds to the nucleotide binding domain-1 (NBD-1) present on CFTR, leading to a rapid, reversible, and voltage-independent inhibition of the channel (33). Cells were then washed once and infected with *M. abscessus* for 3 h at 37°C at a multiplicity of infection (MOI) of 10 in the presence or absence of INH172. Thereafter, extracellular bacilli were killed by a 1-h incubation with 250 $\mu$g/mL amikacin. Finally, cells were lysed with 1% deoxycholate (Sigma), and samples were diluted in phosphate-buffered saline (PBS)-Tween 80 and CFU-quantified by plating bacilli in triplicate on 7H10.

To assess intracellular bacterial growth, dTHP-1 cells and MDM from healthy donors or from CF patients were distributed in 24- or 96-well plates at the concentration of $5 \times 10^5$ cells/mL or $1 \times 10^6$ cells/mL, respectively. Cells were infected with *M. abscessus*, for 3 h at 37°C at an MOI of 10 in the presence or absence of 10 $\mu$M INH172. Thereafter, extracellular bacilli were killed by a 1-h incubation with 250 $\mu$g/mL amikacin. Cells were then washed and incubated with ABL loaded with PA, PI3P, PI5P, S1P, LBPA, or AA used at the ratio of 1:1 (ABL:MDM) for a further 18 h in the presence or absence of INH172. Finally, cells were lysed with 1% deoxycholate (Sigma), and samples were diluted in PBS-Tween 80 and CFU-quantified by plating bacilli in triplicate on 7H10. To evaluate the role of ROS and of phagosome acidification in intracellular bacterial killing, *M. abscessus*-infected cells were treated simultaneously with ABL/PI5P with either polyethylene glycol (PEG)-catalase (100 U/mL) or concanamycin A (1 nM). In order to evaluate the *in vitro* efficacy of a combined therapy on extracellular and intracellular mycobacterial viability, dTHP-1 cells or MDM from CF patients were infected with *M. abscessus* at an MOI of 10 for 3 h at 37°C. Cells were then stimulated, in presence or absence of INH172, with ABL/PI5P and/or 4 $\mu$g/mL amikacin (AMK) for 18 h. Both extracellular and intracellular bacterial growth were assessed by CFU assay.

**Fluorometric analysis.** Phagosome acidification was assessed using the fluorescent probe LysoSensor green DND 189 (Molecular Probes) (34), which measures the pH of acidic organelles, such as phagolysosomes. Briefly, dTHP-1 cells were pretreated or not treated for 1 h with 10 $\mu$M INH172 and then infected with *M. abscessus* for 3 h at 37°C at an MOI of 10 in the presence or absence of 10 $\mu$M INH172. Cells were then washed and incubated for a further 3 h with ABL carrying PA, PI3P, and PI5P, added at the ratio of 1 to 1 (ABL:MDM) in the presence or absence of INH172. Cells were stained for 15 min at 37°C with 1 $\mu$M LysoSensor green DND 189. A pH calibration curve was obtained by incubating macrophages in calibration buffers at pH 4.5, 5.5, 6.5, and 7.5 (intracellular pH calibration buffer kit; Molecular Probes) and by labeling cells for 15 min at 37°C with 1 $\mu$M LysoSensor green DND 189, according to the manufacturer's instructions. The intensity of fluorescence was determined at an excitation wavelength of 492 nm and an emission wavelength of 517 nm using a Varioskan LUX multimode microplate reader (Thermo Fisher Scientific).

ROS generation was analyzed by loading dTHP-1 cells with the fluorescent indicator 20,70-dichloro-fluorescein diacetate (DCF) (Molecular Probes), used at a concentration of 10 $\mu$M, for 40 min at 37°C in the dark. Thereafter, dTHP-1 cells were pretreated or not treated for 1 h with 10 $\mu$M INH172 and then infected with *M. abscessus* for 3 h at 37°C at an MOI of 10 in the presence or absence of 10 $\mu$M INH172. Cells were then washed and incubated for a further 3 h with ABL carrying PA, PI3P, and PI5P, added at the ratio of 1 to 1 (ABL:MDM), in the presence or absence of INH172. The production of ROS was evaluated by fluorometry using a Varioskan LUX multimode microplate reader (Thermo Fisher Scientific) and setting the excitation and emission wavelengths at 488 nm and 530 nm, respectively.

**Mouse model of chronic infection.** The agar bead method was used to establish a stable infection in mice and to faithfully reproduce a chronic infection. In detail, *M. abscessus* colonies from 7H10 plates were grown for 2 days (to reach the exponential phase) in 20 mL of Middlebrook 7H9 broth. A bacterial suspension reaching an optical density (OD) of 15 was used for bead preparation. Then 50 mL of white heavy mineral oil and 25 mL of Trypticase soy agar (TSA) were added to the bacterial suspension and mixed at medium speed with a magnet on a stirrer to generate agar beads of the correct size (100 to 200 $\mu$m). Agar bead preparations were stored at 4°C for no more than a week, as after this time, the number of viable bacteria included within the agar beads decreases. Fresh preparations were performed every time. Embedding *M. abscessus* within agar beads and using intratracheal injection physically retained the bacteria in the bronchial airways and provided microanaerobic/anaerobic conditions that allow bacteria to grow in microcolonies (14). For the inoculum, mice were anesthetized, the trachea was exposed and intubated, and 50 $\mu$L of bead suspension ($1 \times 10^5$ CFU) was injected before closing the incision with suture clips. Control mice were intratracheally inoculated with the same volume of empty bead suspension. Agar bead sizes smaller than 100 to 200 $\mu$m could be easily cleared by mice, while sizes larger than 100 to 200 $\mu$m may not reach the lung during the intratracheal infection. Starting from day 7 after infection and for a further 4 weeks, WT and CF mice were treated with intranasal (i.n.) inoculation of $10^5$ ABL loaded with PA, PI3P, or PI5P suspended in saline solution, given alone or in combination with 100 mg/kg or 200 mg/kg amikacin (AMK), administered by intraperitoneal (i.p.) injection. In particular, mice were treated three times a week for 8, 17, and 29 days and were monitored daily for changes in body weight, appetite, and hair coat; at fixed time points from infection (15, 24, and 36 days)

mice were killed by carbon dioxide euthanasia, and total lung CFU were determined by using a viable count assay with an *M. abscessus* limit of detection of $\geq 10^2$ CFU.

**Evaluation of the *in vivo* inflammatory response.** BALF was obtained as previously described (35), total cells present in the BALF were counted, and a differential cell count was performed on cytospins stained with Diff Quick. Lungs were recovered and homogenized. BALF and lung serial dilutions were plated on 7H10 plates for CFU counting. The lung supernatants were stored at −80℃ for cytokine and chemokine determination. Mouse custom ProcartaPlex 9-plex (Invitrogen, Thermo Fisher Scientific, Waltham, MA, USA) was used for quantification of lung cytokines and chemokines, and values were normalized at 2,500 $\mu$g/mL of quantified proteins in lung supernatants.

**Statistics.** Comparison between groups was performed using Student's *t* test, as appropriate for normally distributed data. The Wilcoxon rank sum test or Mann-Whitney test was performed for data that were not normally distributed.

**Ethics statement.** Buffy coats from anonymized healthy donors, who gave their written informed consent to donate the nonclinically usable components of their blood for scientific research, were obtained from the Blood Transfusion Unit of "Policlinico Tor Vergata" in Rome, Italy (ethics approval no. 16/2020).

Cystic fibrosis patients, giving their (or parental) informed consent to participate in the study, were enrolled at "Bambino Gesù" Children's Hospital in Rome, Italy, after having received detailed information on the scope and objectives of the study by medical personnel, who explained the patient information leaflet (ethics approval no. 738/2017 of "Bambino Gesù" Children's Hospital, Rome, Italy).

Animal studies were approved by the Italian Ministry of Health guidelines for the use and care of experimental animals and registered by San Raffaele Scientific Institute Institutional Animal Care and Use Committee (IACUC no. 966, approved in March 2019).

**Data availability.** All data are available in the main text or the supplemental material.

## SUPPLEMENTAL MATERIAL

Supplemental material is available online only.

**SUPPLEMENTAL FILE 1**, PDF file, 2.3 MB.

## ACKNOWLEDGMENTS

We thank Carla Viscomi, head nurse of the Cystic Fibrosis Unit, Pediatric Hospital "Bambino Gesù" in Rome, Italy, for her important support in the management and collection of clinical samples from patients and Matteo M. Naldini for English revision of the manuscript.

Conceptualization: M.F., D.M.C., V.L.; methodology: N.P., C.R., T.O., M.R., N.I.L., F.C., F.D.S., M.M.D.A., L.H.D.A.; investigation: N.P., C.R., T.O., M.R., F.D.S., N.I.L.; visualization: N.P., C.R., M.M.D.A., L.H.D.A.; funding acquisition: M.F., D.M.C.; project administration: M.F., D.M.C., V.L.; supervision: M.F., D.M.C., V.L.; resources: F.C., V.L.; writing, original draft: M.F., D.M.C.; writing, review and editing: M.F., D.M.C., N.P., C.R., M.M.D.A.

This study was financially supported by the Italian Cystic Fibrosis Foundation (FFC no. 16/2018 and FFC no. 21/2019), Italian Ministry of Defence (no. a2018.092), and FISM-Fondazione Italiana Sclerosi Multipla (2016/R/22) and financed or cofinanced with "5 per mille" public funding.

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
