## [Reviewer comments · Microbiology Spectrum]

Microbiology Spectrum

Combined host- and pathogen-directed therapy for the control of *Mycobacterium abscessus* infection

Noemi POERIO, Camilla Riva, Tommaso Olimpieri, Marco Rossi, Nicola Loré, Federica De Santis, Lucia Henrici De Angelis, Fabiana Ciciriello, Marco D'Andrea, Vincenzina Lucidi, Daniela Cirillo, and Maurizio Fraziano

Corresponding Author(s): Maurizio Fraziano, University of Rome Tor Vergata

Review Timeline:

Submission Date:	December 7, 2021
Editorial Decision:	December 21, 2021
Revision Received:	January 3, 2022
Accepted:	January 7, 2022

Editor: Paolo Visca

Reviewer(s): The reviewers have opted to remain anonymous.

Transaction Report:

DOI: <https://doi.org/10.1128/spectrum.02546-21>

December 21, 2021

Prof. Maurizio Fraziano
University of Rome Tor Vergata
Biology
via della ricerca scientifica, 1
Rome 00133
Italy

Re: Spectrum02546-21 (Combined host- and pathogen-directed therapy for the control of *Mycobacterium abscessus* infection)

Dear Prof. Maurizio Fraziano:

Thank you for submitting your manuscript to Microbiology Spectrum. Your manuscript has now been reviewed by two experts who provided positive comments and asked for few minor revisions. Based in their comments, I would be glad to recommend acceptance of your manuscript after revision. When submitting the revised version of your paper, please provide (1) point-by-point responses to the issues raised by the reviewers as file type "Response to Reviewers," not in your cover letter, and (2) a PDF file that indicates the changes from the original submission (by highlighting or underlining the changes) as file type "Marked Up Manuscript - For Review Only". Please use this link to submit your revised manuscript - we strongly recommend that you submit your paper within the next 60 days or reach out to me. Detailed instructions on submitting your revised paper are below.

Link Not Available

Sincerely,

Paolo Visca

Journals Department
Reviewer comments:

Reviewer #1 (Comments for the Author):

The increase of nontuberculous mycobacteria infections and the lack of an adequate antibiotic therapy represent a major health problem. This manuscript represents an innovative approach to fight this pathogen. The authors validated their approach in vivo also in a CF murine model; the results of the study are supported by experimental data. The manuscript is well written and technically sound. I would like to recommend minor revisions and suggestions.

Please, check the English language style through the full manuscript because there are some typos (e.g.: singular/plural, etc).

Check the italics (species, in vivo, in vitro, etc). Furthermore, the first time that one species is introduced, has to be written also in extenso (e.g.: *Mycobacterium tuberculosis*).

Sometimes the sentences are very long and difficult to read; I suggest improving the text, following this advice (lines 87-91: sentence too long, furthermore, there is "evaluated" twice; lines 129-134.

Results

Lines 138-167: In order to more easily understand the results, I think that in the manuscript the same indication regarding the days of treatment (of infection) in which the BALF was performed should be indicated both in the text and in the Figures. For example, in line 144 "day 15, 24 and 36 from infection" correspond to 8, 17, 29 days of treatment. I think that it could generate confusion in the reader. Interesting results were obtained but maybe in this way they could be more understandable.

Lines 169-197: The treatment is explained well in Material and methods, but I think it is important to remind the conditions at the beginning of this paragraph. The treatment includes a combination of liposomes (intra-nasal administration) and amikacin (ip administration).

Minor revisions

Line 41: BALF: bronchoalveolar lavage fluid. Please write in extenso and put the abbreviation in brackets.

Lines 107-109: explain better the differences among the results showed in Fig. 2S and Fig. 1A.

Lines 123-124: you have already introduced the abbreviation CF; so, you can write "CF patients".

Lines 130-131: responsible for...

Line 246: proinflammatory

Line 277: these?

Line 320: were instead of was.

Line 413: -80{degree sign}C

Please change "u" with " " in ug/ml.

Fig. 1 B and C. Please explain better in the agenda the differences among these figures. I understood the figures from the main text (lines: 114-117) but not from the legenda. It is the same for Fig. S2 B and C.

Improve Figure 3 C.

Reviewer #2 (Comments for the Author):

General comment

The manuscript of Poerio N. et al is interesting. The Authors report data on ex vivo and in vivo activity of a novel therapeutic regimen based on bioactive liposomes plus amikacin against Mycobacterium abscessus infection. Experimental design is appropriate, and results clearly presented.

Specific comments.

Page 2, line 35: "rifamycins", instead of "rifampycins"

Page 2, lines 38, 39, 41: indicate the meaning of CFTR, ROS, BALF

Page 5, line 86: indicate the meaning of MDR

Page 5, line 104: in the Supplementary information, it would be interesting to add a drawing or a microphotography of the liposomes, particularly of asymmetric liposomes mentioned in Materials and methods (page 15, line 295)

Page 6, line 109: add some information here and/or in Materials and methods on the CFTR inhibitor INH172 (page 17, line 352).

Page 6, line 120: indicate the meaning of ROS

Page 7, line 141-142: give some details here or in Materials and methods (page 17, lines 332-333) on the agar bead method used to establish the chronic lung infection

Page 7, line 150: "almost eradicated" is not correct. The authors have to say that in 3 out of 12 mice the total lung CFU was below the limit of detection of the viable count assay (indicate this limit in Materials and methods).

Page 8, line 156: indicate the meaning of KC.

Page 11, line 217: "moxifloxacin", not "moxifloxacina"

Page 12, line 235-236: "Klebsiella", instead of "K."; "Acinetobacter", instead of "A."; "Staphylococcus", instead of "S.". Indicate the meaning of BAL.

Page 16, line 319: indicate the meaning of M-CSF

Page 17, line 342: give some experimental details on "gut corrected"

Page 18, line 370: indicate the meaning of "MOI".

Page 20, line 409: I would prefer "mice were killed by carbon dioxide euthanasia, and total lung CFUs determined" instead of "mice were euthanized by CO2 asphyxiation".

References: the names of the bacterial species should be italicized (pages 22-27)

Page 28, line 537: indicate the meaning of "MDM" and "HD".

Page 28, line 556: explain the effect of Concanamycin A on phagolysosome acidification.

Page 30, line 589: "amikacin", instead of "AMK". I also suggest using "amikacin" instead of "Amikacin" throughout the manuscript, wherever appropriate.

Supplementary Table 1. Indicate the meaning of FEV-1, and explain while the age of each patient was indicated as age range.

Staff Comments:

Preparing Revision Guidelines

Please return the manuscript within 60 days; if you cannot complete the modification within this time period, please contact me. If you do not wish to modify the manuscript and prefer to submit it to another journal, please notify me of your decision immediately so that the manuscript may be formally withdrawn from consideration by Microbiology Spectrum.

The increase of nontuberculous mycobacteria infections and the lack of an adequate antibiotic therapy represent a major health problem. This manuscript represents an innovative approach to fight this pathogen.

The authors validated their approach *in vivo* also in a CF murine model; the results of the study are supported by experimental data. The manuscript is well written and technically sound.

I would like to recommend it for the publication after minor revisions and suggestions.

Please, check the English language style through the full manuscript because there are some typos (e.g.: singular/plural, etc).

Check the italics (species, *in vivo*, *in vitro*, etc). Furthermore, the first time that one species is introduced, has to be written also in extenso (e.g.: *Mycobacterium tuberculosis*).

Sometimes the sentences are very long and difficult to read; I suggest improving the text, following this advice (lines 87-91: sentence too long, furthermore, there is “evaluated” twice; lines 129-134).

Results

Lines 138-167: In order to more easily understand the results, I think that in the manuscript the same indication regarding the days of treatment (of infection) in which the BALF was performed should be indicated both in the text and in the Figures. For example, in line 144 “day 15, 24 and 36 from infection” correspond to 8, 17, 29 days of treatment. I think that it could generate confusion in the reader. Interesting results were obtained but maybe in this way they could be more understandable.

Lines 169-197: The treatment is explained well in Material and methods, but I think it is important to remind the conditions at the beginning of this paragraph. The treatment includes a combination of liposomes (intranasal administration) and amikacin (ip administration).

Minor revisions

Line 41: BALF: bronchoalveolar lavage fluid. Please write in extenso and put the abbreviation in brackets.

Lines 107-109: explain better the differences among the results showed in Fig. 2S and Fig. 1A.

Lines 123-124: you have already introduced the abbreviation CF; so, you can write “CF patients”.

Lines 130-131: responsible for...

Line 246: proinflammatory

Line 277: these?

Line 320: were instead of was.

Line 413: -80°C

Please change “u” with “μ” in ug/ml.

Fig. 1 B and C. Please explain better in the agenda the differences among these figures. I understood the figures from the main text (lines: 114-117) but not from the legenda. It is the same for Fig. S2 B and C.

Improve Figure 3 C.

Point-by-point reply to reviewers' comments

Reviewer #1

Reviewer #1. The increase of nontuberculous mycobacteria infections and the lack of an adequate antibiotic therapy represent a major health problem. This manuscript represents an innovative approach to fight this pathogen. The authors validated their approach *in vivo* also in a CF murine model; the results of the study are supported by experimental data. The manuscript is well written and technically sound. I would like to recommend minor revisions and suggestions.

Authors. We would like to thank the reviewer for the positive evaluation and constructive comments that have been used to improve the manuscript. Please find below a detailed point-by-point response to all comments.

R#1. Please, check the English language style through the full manuscript because there are some typos (e.g.: singular/plural, etc).

A. Text has been revised and modified where necessary as suggested by the reviewer. All changes were marked in bold red.

R#1. Check the italics (species, *in vivo*, *in vitro*, etc). Furthermore, the first time that one species is introduced, has to be written also in extenso (e.g.: *Mycobacterium tuberculosis*).

A. Text has been revised and modified where necessary as suggested by the reviewer. All changes were marked in bold red.

R#1. Sometimes the sentences are very long and difficult to read; I suggest improving the text, following this advice (lines 87-91: sentence too long, furthermore, there is "evaluated" twice; lines 129-134).

A. The sentences have been modified as suggested in order to improve their clearness and marked in bold red.

Results

R#1. Lines 138-167: In order to more easily understand the results, I think that in the manuscript the same indication regarding the days of treatment (of infection) in which the BALF was performed should be indicated both in the text and in the Figures. For example, in line 144 "day 15, 24 and 36 from infection" correspond to 8, 17, 29 days of treatment. I think that it could generate confusion in the reader. Interesting results were obtained but maybe in this way they could be more understandable.

A. We agree with the reviewer's suggestion. The text has been modified according to the reviewer indication and marked in bold red.

R#1. Lines 169-197: The treatment is explained well in Material and methods, but I think it is important to remind the conditions at the beginning of this paragraph. The treatment includes a combination of liposomes (intra-nasal administration) and amikacin (ip administration).

A. We thank the reviewer for the suggestion and the text has been modified according to the reviewer's indication and marked in bold red.

Minor revisions

R#1. Line 41: BALF: bronchoalveolar lavage fluid. Please write in extenso and put the abbreviation in brackets.

A. The text has been modified as requested and marked in bold red.

R#1. Lines 107-109: explain better the differences among the results showed in Fig. 2S and Fig. 1A.

A. The sentence has been modified as suggested by the reviewer in order to improve its comprehensibility and has been marked in bold red.

R#1. Lines 123-124: you have already introduced the abbreviation CF; so, you can write "CF patients".

A. The text has been modified as requested and marked in bold red.

R#1. Lines 130-131: responsible for...

A. The text has been modified as requested and marked in bold red.

R#1. Line 246: proinflammatory

A. The text has been modified as requested and marked in bold red.

R#1. Line 277: these?

A. The text has been modified as requested and marked in bold red.

R#1. Line 320: were instead of was.

A. The text has been modified as requested and marked in bold red.

R#1. Line 413: -80{degree sign}C

A. The text has been modified as requested and marked in bold red.

R#1. Please change "u" with "μ" in ug/ml.

A. The text has been modified as requested and marked in bold red.

R#1. Fig. 1 B and C. Please explain better in the agenda the differences among these figures. I understood the figures from the main text (lines: 114-117) but not from the legenda. It is the same for Fig. S2 B and C.

A. The figure legends have been modified as suggested in order to improve their clearness and have been marked in bold red.

R#1. Improve Figure 3 C.

A. We have proceeded to improve the clearness and the quality of the figure as suggested by the reviewer.

Reviewer #2:

General comments

Reviewer #2. The manuscript of Poerio N. et al is interesting. The Authors report data on *ex vivo* and *in vivo* activity of a novel therapeutic regimen based on bioactive liposomes plus amikacin against *Mycobacterium abscessus* infection. Experimental design is appropriate, and results clearly presented.

A. We would like to thank the reviewer for the appreciation of our work and for his/ her valuable and insightful comments. We have taken the suggestions on board to improve and clarify the manuscript. Please find below a detailed point-by-point response to all comments.

Specific comments.

R#2. Page 2, line 35: "rifamycins", instead of "rifampycins"

A. The text has been modified as requested and marked in bold red.

R#2. Page 2, lines 38, 39, 41: indicate the meaning of CFTR, ROS, BALF

A. The text has been modified as requested and marked in bold red.

R#2. Page 5, line 86: indicate the meaning of MDR

A. The text has been modified as requested and marked in bold red.

R#2. Page 5, line 104: in the Supplementary information, it would be interesting to add a drawing or a microphotography of the liposomes, particularly of asymmetric liposomes mentioned in Materials and methods (page 15, line 295)

A. We agree with the reviewer's suggestion, and we have revised the supplementary information by adding a panel with a representative drawing of asymmetric liposomes to the supplementary figure 1 (Fig S1A). The main text has been modified accordingly and marked in bold red.

R#2. Page 6, line 109: add some information here and/or in Materials and methods on the CFTR inhibitor INH172 (page 17, line 352).

A. As requested by the reviewer we have proceeded to modify the text by adding additional information, and the relative bibliographic citation, regarding the mechanisms of action of CFTR inhibitor INH172 in the "materials and methods" section. The modifications have been indicated in the text in bold red.

R#2. Page 6, line 120: indicate the meaning of ROS

A. The text has been modified as requested and marked in bold red.

R#2. Page 7, line 141-142: give some details here or in Materials and methods (page 17, lines 332-333) on the agar bead method used to establish the chronic lung infection

A. As per reviewer's request, we have proceeded to modify the text by adding details about the agar bead method used in the "materials and methods" section. The modifications have been indicated in the text in bold red.

R#2. Page 7, line 150: "almost eradicated" is not correct. The authors have to say that in 3 out of 12 mice the total lung CFU was below the limit of detection of the viable count assay (indicate this limit in Materials and methods).

A. We apologize to the reviewer for using inappropriate language in expressing the concept. In particular, we wanted to convey the message that the pathogen was not detectable by performing the vitality count assay and by using a detection limit of *Mycobacterium abscessus* greater than or equal to 100 CFU. Thus, as requested by the reviewer, we have proceeded to correct the sentence and improved the "materials and methods" section by adding the limit of detection of the viable count assay. Changes in the text have been marked in bold red.

R#2. Page 8, line 156: indicate the meaning of KC.

A. The text has been modified as requested and marked in bold red.

R#2. Page 11, line 217: "moxifloxacin", not "moxifloxacina"

A. The text has been modified as requested and marked in bold red.

R#2. Page 12, line 235-236: "Klebsiella", instead of "K."; "Acinetobacter", instead of "A."; "Staphylococcus", instead of "S.". Indicate the meaning of BAL.

A. The text has been modified as requested and marked in bold red.

R#2. Page 16, line 319: indicate the meaning of M-CSF

A. The text has been modified as requested and marked in bold red.

R#2. Page 17, line 342: give some experimental details on "gut corrected"

A. As per reviewer request, we have inserted in the text a more specific description about the CF animal model used, and in particular about the "gut correction". The modifications and the related reference have been indicated in the text in bold red.

R#2. Page 18, line 370: indicate the meaning of "MOI".

A. The text has been modified as requested and marked in bold red.

R#2. Page 20, line 409: I would prefer "mice were killed by carbon dioxide euthanasia, and total lung CFUs determined" instead of "mice were euthanized by CO2 asphyxiation".

A. The text has been modified as suggested and marked in bold red.

R#2. References: the names of the bacterial species should be italicized (pages 22-27)

A. The text has been modified as requested and marked in bold red.

R#2. Page 28, line 537: indicate the meaning of "MDM" and "HD".

A. The text has been modified as requested and marked in bold red.

R#2. Page 28, line 556: explain the effect of Concanamycin A on phagolysosome acidification.

A. As requested by the reviewer we have proceeded to improve the text by adding the mechanism of action of Concanamycin A on phagolysosome acidification inhibition. The modification has been marked in bold red.

R#2. Page 30, line 589: "amikacin", instead of "AMK". I also suggest using "amikacin" instead of "Amikacin" throughout the manuscript, wherever appropriate.

A. We agree with the reviewer's suggestion. The text has been modified as requested and marked in bold red.

R#2. Supplementary Table 1. Indicate the meaning of FEV-1 and explain while the age of each patient was indicated as age range.

A. The text has been modified as requested and marked in bold red.

January 7, 2022

Prof. Maurizio Fraziano
University of Rome Tor Vergata
Biology
via della ricerca scientifica, 1
Rome 00133
Italy

Re: Spectrum02546-21R1 (Combined host- and pathogen-directed therapy for the control of *Mycobacterium abscessus* infection)

Dear Prof. Maurizio Fraziano:

I greatly appreciated your smart drug delivery approach and its efficacy in treatment of Mab infection in the CF mouse model. I hope your research will move forward in the preclinical stage.

Your revised manuscript has now been accepted, and I am forwarding it to the ASM Journals Department for publication. You will be notified when your proofs are ready to be viewed.

Sincerely,

Paolo Visca
Editor, Microbiology Spectrum
